# Surface Functionalization of Polyester Textiles for Antibacterial and Antioxidant Properties

**DOI:** 10.3390/polym14245512

**Published:** 2022-12-16

**Authors:** Esam S. Allehyani

**Affiliations:** Department of Chemistry, University College in Al-Jamoum, Umm Al-Qura University, Makkah 24211, Saudi Arabia; esklehyani@uqu.edu.sa

**Keywords:** ethylenediamine, polyester, tensile strength, antioxidant, antibacterial

## Abstract

One of the recommendations for future textile development is the modification of textiles to produce materials for human performance (sports, medical, and protective). In the current work, modifying a polyester surface with silver nanoparticles improved antioxidant and antibacterial protection. For this purpose, ethylenediamine aminolysis was utilized as ligands to fabricate polyester textiles, trapping silver ions to further reduce silver nanoparticles (AgNPs). Dopamine (PDA) was used to provide antibacterial and antioxidant properties to the polyester textile by converting silver ions into AgNPs through its phenolic hydroxyl groups. Pristine polyester, polyester treated with ethylenediamine, and PDA-coated AgNP-loaded polyester ethylenediamine were characterized using SEM, EDX, FTIR, TGA, and tensile strength. The antibacterial properties against *Staphylococcus aureus* and *Escherichia coli* were examined through the broth test. PDA-AgNPs composite nanocoating exhibited improved tensile strength and antibacterial and antioxidant properties, demonstrating that polyester with a PDA-AgNPs overlay may be used for long-term biomedical textiles.

## 1. Introduction

Textiles may provide an ideal microenvironment for microorganism growth by providing adequate nutrition, moisture, and oxygen. To create technological materials and prevent the degradation caused by microorganisms, functionalizing textiles using antimicrobial finishing compounds has been widely employed. These textiles may be used in various applications to limit microbial growth and, consequently, unpleasant odors, stains, infections, a decline in the mechanical qualities of the textile, and cross-contamination [1]. Microorganisms may persist for lengthy periods on inanimate “touch” surfaces. This is especially problematic in health care, as patient immunity is in danger of infection. Touch surfaces in hospital rooms might act as a source or reservoir for bacterial growth [2]. Polyester fabrics with several applications, especially in hygiene and health, are receiving more attention [3]. The main disadvantage is the tendency for pathogenic microorganisms to grow on polyester textiles; as a result, research on antibacterial materials is significant [4]. Given the growing variety of bacteria strains immune to antibiotics and silver’s minimal toxicity, using restricted amounts of silver as antibiotic agents for antibacterial materials is a promising method for potential implementation [5]. Nanocomposite-based coatings have the potential to combat pathogenic microorganisms by interacting directly with the bacterial cell wall, inhibiting biofilm formation, activating intrinsic and adaptive host immune responses, producing reactive oxygen species (ROS), and interacting with intracellular components [6]. Different metal nanoparticles (NPs) have attracted a great deal of interest as potential antibacterial agents. The antibacterial action of silver-containing products is thought to be derived from the fact that silver ions demonstrate a wide range of bactericidal activity against various pathogenic microorganisms. These characteristics make silver nanoparticles (AgNPs) an innovative iteration of medical materials [7]. These are widely utilized nowadays in a variety of industries, including the food industry, cosmetics, coatings, sunscreen, medical equipment, clothes, and women’s hygiene products [8,9]. This increasing utilization of AgNPs-improved commodities could increase toxicity, impacting both the environment and living creatures if substantial quantities of AgNPs or Ag+ ions are released into the environment, in addition to the impacts of their long- or short-term exposure on living creatures and/or human physiology [10,11]. Compounds containing silver release silver ions, which react with the thiols in proteins essential to microbial survival and are primarily responsible for the antibacterial activity of silver. These interactions prevent cells from respiring normally and cause cell death. The existence of halide ions, which commonly produce a reaction when put together with silver ions, may affect or reduce its antibacterial action [9]. As a result, free silver ions have a very limited antibacterial action when utilized alone. Silver nanoparticles (AgNPs) and composites incorporating AgNPs have recently been created to solve this problem. Silver ions are progressively liberated from these materials, resulting in antibacterial action [10,11]. When AgNPs are synthesized ex-situ and physically introduced into textiles, or when utilizing traditional physicochemical techniques, AgNPs typically exhibit poor energetic adhesion onto fibers. In-situ AgNPs synthesis in textile fibers is being explored to solve this persistent adhesion issue [12,13]. A further scientific examination is needed regarding the in-situ production of AgNPs in polyester textiles with various surface characteristics, such as wettability, ruggedness, and permeability. The treatment with ethylene diamine has the advantage of activating the PET fabric with amidoethylene amine, which would render the fabric basicity and/or crosslinking the possibility with adjacent chains. Furthermore, mimicking the structural amidoethylene amine unit with the replaced ethylene glycol unit in the PET fabric could lead to activated fabric with interesting potential characteristics.

## 2. Methods and Materials

Silver nitrate, ethylenediamine (EDA), dopamine, methanol, and Tris-hydrochloride were procured from Sigma-Aldrich (St. Louis, MO, USA). Polyester woven fabrics (195 g/m^2^) (poly(ethylene terephthalate)) were acquired through Misr El-Mahalla Co., Egypt.

### 2.1. Polyester Treatment

A particular weight of polyester was treated using 12% ethylenediamine in methanol at a liquor-to-goods ratio of 30:1 for 30 min at 60 °C. The treated polyester underwent a careful overnight immersion in 50 mM silver nitrate, then washing in water and meticulous overnight immersion in 6 mg/mL dopamine prepared in 10 mM Tris HCl, pH 8.5, followed by water washing and air drying [13].

### 2.2. Characterization

To study the morphological characters, measurements of nanocomposites were made via scanning electron microscopy (SEM) (Quanta FEG 250, FEI Co., operating at 20 kV, Hillsboro, OR, USA) and energy-dispersive X-ray spectroscopy (EDX). The polyester fabric was covered with gold, secured with Quanta holding stubs, and inspected in a vacuum. Attenuated Total Reflectance Fourier-Transform Infrared Spectroscopy was used to investigate the chemical makeup of modified polyester (ATR–FTIR, PerkinElmer Spectrum 100). Tensile strength and elongation at break tests were run using a Galdabini Quasar 50 kN Universal Testing Machine equipped with a 300 mm/min crosshead speed and a 25 kN load cell. The tensile strength was computed using ISO 2307:2010. The specimens underwent thermogravimetric analysis (TGA) with a heating rate of 10 °C min^−1^ using a Shimadzu DTA/TGA-50, Japan.

### 2.3. Antioxidant Activity

#### 2.3.1. Ferric Reducing Antioxidant Power Assay (FRAP)

Benzie and Strain’s (1996) method was used with some modifications [14]. The basic concept behind this approach is reducing a ferric 2,4,6-tripyridyl-s-triazine complex (Fe^3+^-TPTZ) to its ferrous, colored form (Fe^2+^-TPTZ) in the presence of antioxidants. The FRAP reagent contains 25 mL of 0.3 M sodium acetate buffer, pH 3.6, 2.5 mL of 5 mM FeCl_3_, and 2.5 mL of a 10 mM TPTZ (2,4,6-tripyridyl-s-triazine) solution in 40 mM HCl. It was prepared daily and heated to 37 °C. Aliquots of 30 mg of the sample were combined with 1.8 mL of FRAP reagent. After 30 min of incubation at 37 °C against the sugar analog, the reaction mixture’s absorbance was measured spectrophotometrically at 593 nm.

#### 2.3.2. DPPH (2,2-diphenyl-1-picrylhydrazyl) Assay

A 10 mL solution containing 0.1 mM DPPH was added to 50 mg of each sample, which was then incubated at room temperature in the dark for 1, 2, and 4 h to evaluate the polyester’s antioxidant activity. The absorbance of the solutions at predefined time intervals at a wavelength of 515 nm was measured using a UV-visible spectrophotometer. The following equation was used to determine the antioxidant efficiency [15].
(1)$$\mathrm{Antioxidant}\;\mathrm{efficiency}\;\left(\%\right)=\frac{\left(A\;-\;A0\right)}{A}\times 100$$
where A and A0 are the absorbance of DPPH solutions without and with polyester at 515 nm.

### 2.4. Antimicrobial Study

According to Nada et al. [16], the treated polyester textiles’ antibacterial potentials were evaluated in the context of two bacterial strains. The Department of Microbiology at the Agriculture College at Cairo University generously provided the Gram-positive *Staphylococcus aureus* ATCC 25923 and Gram-negative *E. coli* O157:H7 EHEC1-2 93111 strains. The bacteria were stored in glycerol at a concentration of 25% (*w*/*v*) and at a temperature of −20 °C.

### 2.5. Broth Assay

Conserved bacterial cultures were added to a broth made of tryptone soy and incubated for 24 h at 37 °C before being subcultured for an additional day at that same temperature. The following criteria were used to evaluate polyester textiles’ antibacterial properties: The fabric samples were autoclaved for 15 min at 121 °C to sterilize them. After weighing out 0.1 g of every sample, we distributed it evenly among the wells of a 24-well microplate, added 2 mL of nutrient broth to all, and infected the plate with 20 L of newly stimulated bacterial culture to accomplish an ultimate density of 106 CFU/mL. A nutrient broth infected with test microorganisms but without textiles served as the control sample. Three copies of the test were run [17]. Each sample’s bacterial count was assessed using the drop plate technique after 24 h of incubation at 37 °C [18]. Samples with a bactericidal impact were defined as showing no bacterial viability, and specimens with a bacteriostatic impact had no growth or reduction in initial bacterial counts.

## 3. Results and Discussion

In this work, ethylenediamine (EDA) was used to activate polyester fabric (PET), giving it an active amino group that can respond to silver loading. Thus, silver ions can be converted into silver nanoparticles (AgNPs) by dopamine reduction. AgNPs were synthesized on the surface of polyester textiles, as shown in Figure 1. This polyester treatment was achieved via EDA, and the weight increased by 10.8%. After being immersed in silver nitrate and polymerized by dopamine, the uptake of AgNPs was 34%. Considerable attention has been given to antibacterial silver coatings on textiles created by various coating methods. Different methods were utilized for coating the surfaces of textiles. For instance, the PET surface was coated with silver film using high-power impulse magnetron sputtering [19]. Gün Gök et al. [20] used hexamethylenediamine to activate the PET surface for the capture of silver nanoparticles to produce antimicrobial fibers.

Figure 2 and Table 1 illustrate the FTIR of untreated polyester (PET) samples and those treated afterward. The characteristic peaks for C-H aliphatic asymmetric stretching vibration, C-H aromatic, and carbonyl ester vibration were all seen in the untreated PET at 2967 cm^−1^, 3073 cm^−1^, and 1709 cm^−1^, respectively. The ester group’s C-O vibrations were observed between 1010 and 1240 cm^−1^, while the C-H aromatic out-of-plane bending vibrations were observed between 717 and 820 cm^−1^. Similar FTIR data reports on amino polyester can be found in [13,21,22,23]. As seen in Figure 1, after the addition of EDA, the peaks either shifted with increased intensity as a result of band overlap for those due to ester, amino, and amide groups, or broadened with the appearance of new peaks due to N-H amine at 3206 and 3436 cm^−1^ and a shoulder at nearby 1640 cm^−1^ due to carbonyl amide, indicating the successful formation of amino groups (PET-EDA). Once loaded with silver ions, the treated sample underwent dopamine treatment to create PET-EDA-AgNPs-PDA, indicating that the surface was prepared to ligate with them. The FTIR clearly shows that the PDA coating layer’s peaks, particularly those between 955 and 2145 cm^−1^, intersect with those of carboxylate and amino groups and are hence unidentifiable.

Figure 3 shows the architectural surface morphology of the untreated polyester, the effects of the EDA treatment, and the morphological change after treating PET-EDA with silver nitrate-dopamine to create an AgNPs coating. Figure 3a illustrates the PET’s morphology, which has a uniformly flat surface. Following EDA treatment, a considerable variation was seen on the polyester fabric’s surface, as shown in Figure 3b. The ethylenediamine-treated polyester’s (PET-EDA) surface structure appeared to be irregular and rough. In addition, the surface became rough and disorganized after the treatment with silver nitrate-dopamine, as shown in Figure 3c for the PET-EDA-AgNPs-PDA. The EDX confirmed that the AgNPs coating was successful. EDX is a widely accepted method for determining the elemental composition of large samples [9], as X-rays are produced on a surface area around 2 µm deep. Figure 3d and the accompanying table illustrate the silver coating on the polyester fabric, which amounts to 15.41% by weight.

TGA is an appropriate method to determine structural changes in a material’s thermal stability. According to Figure 4, three stages of degradation were seen in all samples. The first occurred was between ambient temperature and 360 °C alongside a mass reduction of around 5% caused by the elimination of absorbed moisture. With an average mass loss of 70%, the second stage of polymer degradation occurred between 360 and 417 °C. The third stage occurred between 490 and 600 °C. The thermal stability variation between specimens found in the second stage up to 417 °C shows that stability rose as follows: PET-EDA > PET-EDA-AgNPs-PDA > Pristine PET. The third stage, which occurs between 490 and 600 °C, exhibited rapid deterioration in the following order: PET-EDA > PET-EDA-AgNPs-PDA > Pristine PET, indicating the effectiveness of the coating.

With the amine functionalization, the polyester fabric should become more hydrophilic after aminolysis. According to the textile industry, more hydrophilic fabrics tend to be less crystalline. As a result, it stands to reason that the tensile strength of the cloth would decline following such chemical treatment [24]. Figure 5 shows that the polyester fabric lost 46% of its tensile strength and 22% of its break elongation after EDA treatment. The treatment reduced the polyester fabric’s elongation at break, which can be attributed to intramolecular and intermolecular hydrogen bonding in terms of the amines. One more possibility is the reduced fiber strength and extensibility from increased surface-level weak areas inside the fiber [25]. The tensile strength of PDA-coated AgNPs-laden PET-EDA was much higher than that of PET-EDA, showing that loaded PDA-coating and AgNPs enhance tensile integrity. The break elongation of PDA-coated samples was higher than that of PET-EDA samples. Several future studies can now be justified by these findings. There was a 76% improvement in tensile strength in composites made of polyester resin or graphene oxide, as reported by Bora et al. [26]. There was a 17% increase in tensile strength in CNT-reinforced polyester composites [27]. According to our finding, PDA acts as an AgNPs capping agent, which increases the fabric’s pliability.

Scientists have developed materials that may offer antioxidant capabilities in response to the growing demand for biomedical textiles. The antioxidant activity was tested using the DPPH method, and the findings are shown in Figure 6. The PET-EDA sample significantly improved as an antioxidant once the period was extended to 4 h, but the PET continued to exhibit very little antioxidant activity. According to reports, amine groups exhibit reducing properties and the potential to scavenge DPPH radicals [28]. Interestingly, using polyester textiles with silver nanoparticles incorporated significantly increased the antioxidant activity since the silver nanoparticles can stabilize free radicals by delocalizing them. Additionally, the PDA coating layer’s presence increased antioxidant activity, demonstrating the efficacy of PDA’s phenolic hydroxyl groups as radical scavengers. AgNPs and phenolics influence antioxidant characteristics, according to reports [29,30,31,32]. As illustrated in Figure 7, a FRAP test was performed to confirm the DPPH assay results by determining the reducing power of the studied materials. PDA-coated AgNPs-loaded PET and PET-EDA were shown to have the highest levels of antioxidant activity, while pure PET exhibited very little. The results of this FRAP test demonstrate the efficacy of the high-powered antioxidant polyester fabric.

All polyester specimens had to be tested by broth assay to determine any promotion in the growth of bacteria or inhibited its spread. Bacterial counts (in CFU/mL) for *Escherichia coli* and *Staphylococcus aureus* in all polyester samples are shown in Figure 8. (PET, PET-EDA, PET-EDA-AgNPs-PDA). Gram-negative bacteria are more resistant to antibiotics due to their thicker cell walls, which is a well-known fact. The potentially dangerous Gram-negative *E. coli* was studied as a model-resistant bacterium utilizing a broth experiment. Bacterial mortality is reduced in Figure 8 due to the presence of the pristine sample. Contrarily, the PET-EDA sample showed significant bioactivity against Gram-positive *St. aureus* and Gram-negative *E. coli.* This finding emphasizes how amine groups break down the cell walls of microbes by snatching away the metal ions that bacteria need to survive. It has been observed that treated polyester has comparable antibacterial potential [33]. Additionally, AgNPs-loaded PET-EDA demonstrated bactericidal activity against *E. coli* while exhibiting bacrestatic activity against *St. aureus*. The antibacterial effect of AgNPs is mediated by breaking microbial cell walls [34,35,36]. The antibacterial effect of silver can primarily be attributed to the fact that silver-containing items give off silver ions when disturbed [37]. These silver ions mix with thiols on proteins and enzymes essential for survival at the microbial level, preventing respiratory functions and, as a result, leading to the cells dying. It was hypothesized that the redox reaction between the immobilized silver ion in the PET-EDA and dopamine would produce PDA, which may serve as a coating layer to increase the fabric’s efficiency and antibacterial capabilities. Bactericidal activity of PDA-coated AgNPs-loaded PET-EDA was seen against all investigated pathogens and bacteria, with the highest antimicrobial activity observed against *E. coli* for instance. Zemljič et al. [38] demonstrated that the PET plastic film was successfully functionalized with chitosan. It was utilized to inhibit *E. coli* and *L. monocytogenes*.

## 4. Conclusions

Modifying PET textiles with ethylenediamine, silver nanoparticles, and dopamine improved the fabric surface. High surface coverage by terminal primary amine groups results from this chemical modification. Amino groups can be introduced to PET fibers to change their hydrophilicity and create sites for immobilizing silver ions, which dopamine can then convert into silver nanoparticles. FTIR, SEM, and EDX characterization techniques were used to confirm the effective production and deposition of silver nanoparticles. Additionally, the loaded AgNPs and PDA-coating improved the tensile strength while exhibiting greater elongation during break than the other option, i.e., polyester treated with only ethylenediamine. This method also revealed antibacterial and antioxidant effects. Due to its antioxidant and antibacterial qualities, PDA-AgNPs nanocoating of polyester is proposed for use in biomedical textiles.

## Figures and Tables

**Figure 1 polymers-14-05512-f001:**
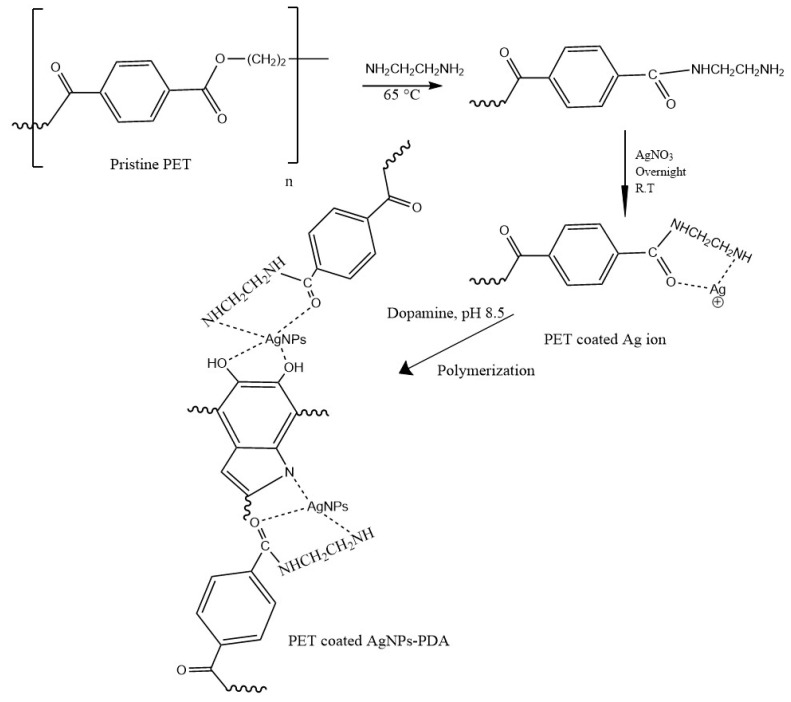
Illustrates the modification of polyester textile.

**Figure 2 polymers-14-05512-f002:**
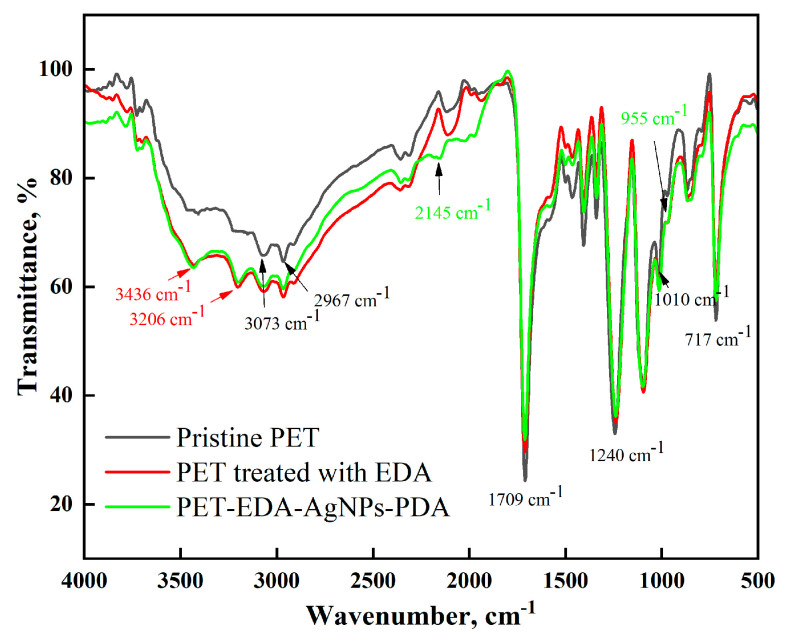
FTIR spectra of polyester before and after modification.

**Figure 3 polymers-14-05512-f003:**
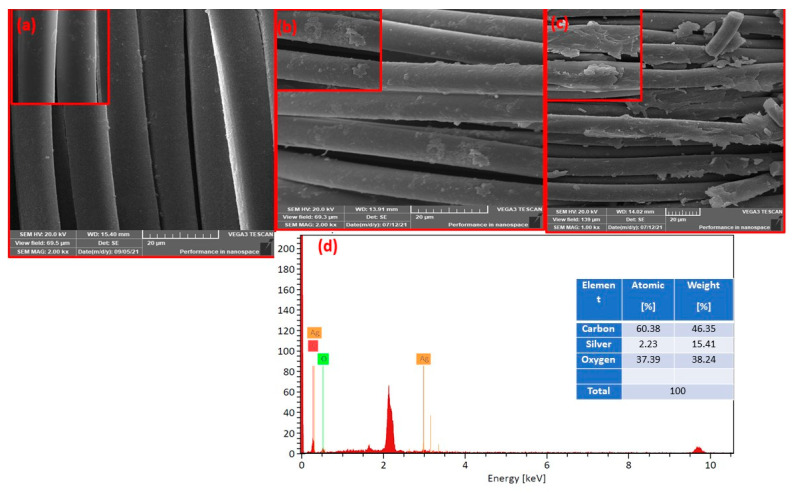
High and low magnification SEM images of (**a**) PET, (**b**) PET-EDA, (**c**) PET-EDA-AgNPs-PDA, and (**d**) SEM–energy-dispersive X-ray (EDX) spectra of PET-EDA-AgNPs-PDA.

**Figure 4 polymers-14-05512-f004:**
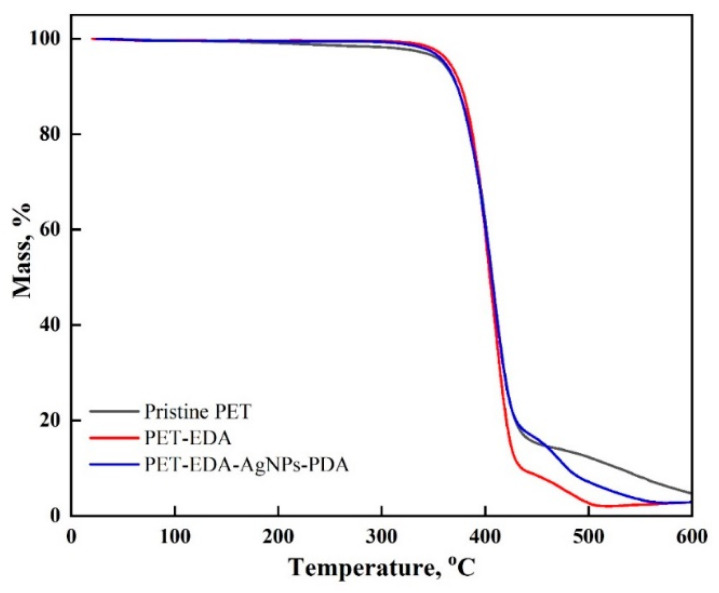
Thermograms of polyester before and after modification.

**Figure 5 polymers-14-05512-f005:**
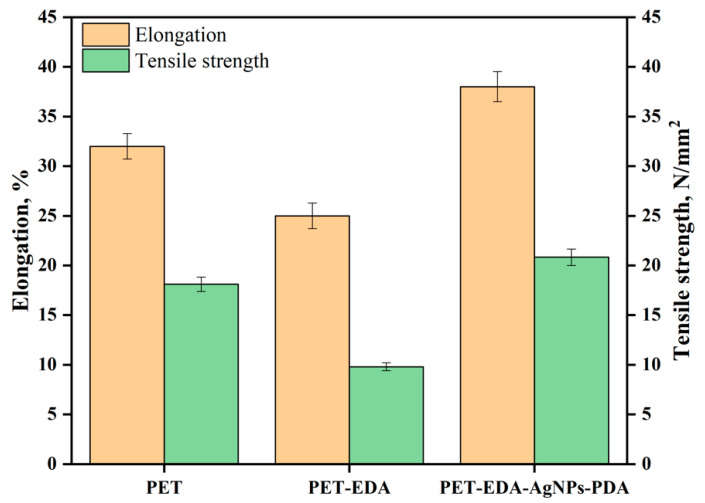
Effect of tensile strength and elongation on polyester before and after modification.

**Figure 6 polymers-14-05512-f006:**
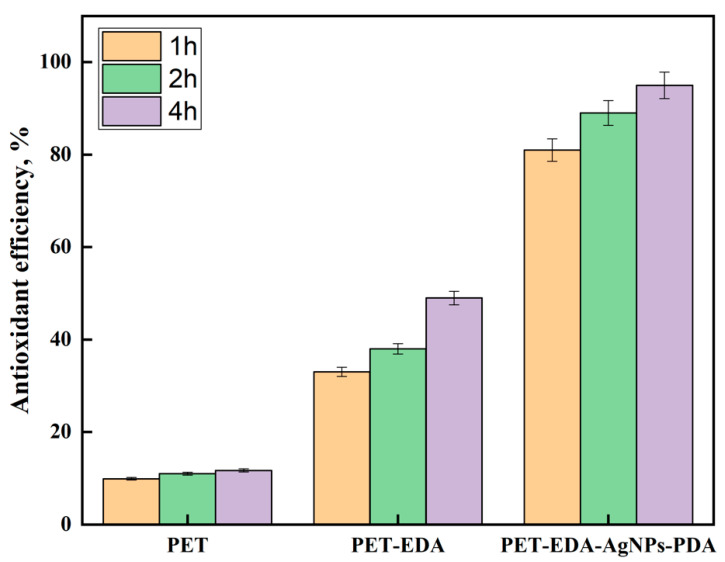
The DPPH antioxidant efficiency at different time of polyester before and after modification.

**Figure 7 polymers-14-05512-f007:**
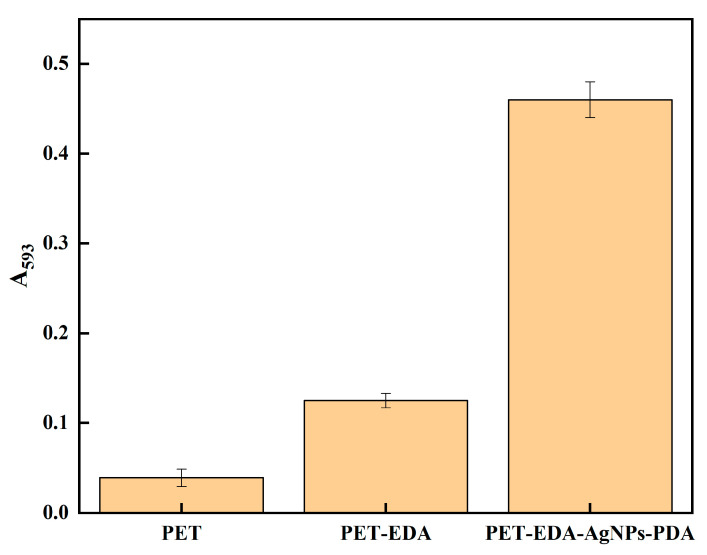
The FRAP reducing power of polyester before and after modification.

**Figure 8 polymers-14-05512-f008:**
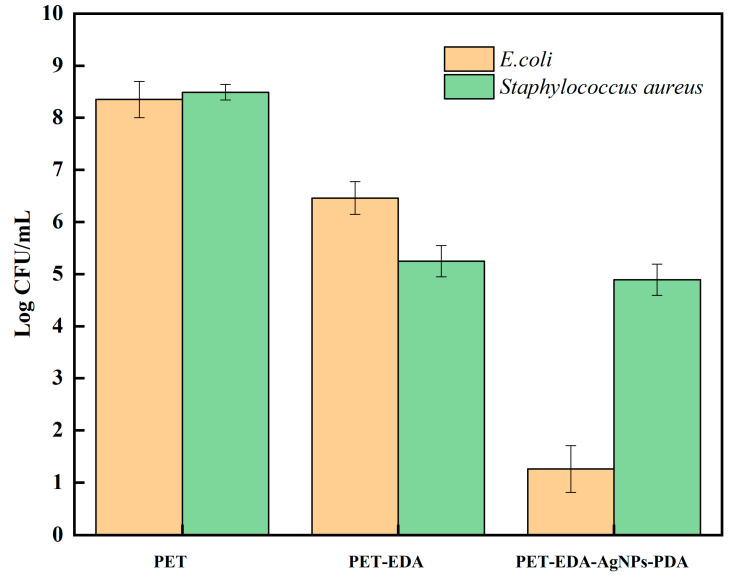
Effect of polyester before and after modification on the microbial growth of *E. coli* and *Staphylococcus aureus*, after incubating for 24 h at 37 °C.

**Table 1 polymers-14-05512-t001:** ATR-FTIR characteristic absorption peaks of PET before and after coating with AgNPs-PDA.

Wavenumber, cm^−1^		
Peak Assignment	PET	PET-EDA-AgNPs-PDA
C–O stretching vibration	1709	1713
C–O ester vibration	1010–1240	1010–1240
C–H aliphatic asymmetric stretching vibration	2967	2973
C–H aromatic asymmetric stretching vibration	3073	3082
C–H out of plane vibration	717–820	717–820
C–N stretching vibration	-	3436, 3206
Carboxylate and amino group stretching vibration	-	955, 2145

## Data Availability

The data presented in this study are available in this same article.

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
