# Peer review of "Surface Functionalization of Polyester Textiles for Antibacterial and Antioxidant Properties"

_polymers, 2022, doi:10.3390/polym14245512_

Round 1

Reviewer 1 Report

The paper must be cardinally improved before will be accepted for publication.

The authors recently published a similar paper where hydrazine was used as PET activator instead ethylenediamine. It is unclear what is advantages and disadvantages of this replacement are.

The problem is not highlighted in the introduction, why the authors decided to functionalize the polyester textiles is hard to understand. To stimulate the interest of the readers it is important to add information on the virucidal potential obtained samples (https://doi.org/10.1016/j.cej.2022.137048).

What type of polyester woven fabric was used exactly? If it is PET why at PET hydrolysis was created two hydroxyl groups? Where on the scheme is dopamine or glucose? The depicted creation of the bonds between oxygen in the carboxylic group and amine in the ethylenediamine is principally impossible!!!!! please carefully check the scheme basing on cited references 16 and 17. In general, the scheme must be strongly corrected, all stages, intermediates, and chemical substances should be written.

Polyester treatment in the methodology was written confusingly and should be rewritten, references to the relevant methodology should be done.

Why does for comparison the fabrication of the silver nanoparticles not conducted for untreated PET?

Please add FTIR results in the form of the Table with signals and assigned groups and compare them to signals presented in references 16 and 17.

Please calculate how many moles and molecules of ethylenediamine were attached to the unit of the surface area and mass of the PET.

Please cite highly relevant papers where similar results were presented.

https://doi.org/10.1016/j.cej.2022.137048

https://doi.org/10.3390/polym14061138

English and typos should be carefully corrected.

Author Response

Thank you very much for your review, please check the attachment.

Reviewer 2 Report

Dear Author,

The manuscript prepared for review presents the functionalization of the surface of polyester textiles in terms of their antibacterial and antioxidant properties. 

The use of silver compounds in the form of nanoparticles is a very well-known and widespread solution, mainly due to its well-known properties.

In the literature you can find works on polyester composites with nanosilver [Pączkowski, P.; Puszka, A.; Miazga-Karska, M.; Ginalska, G.; Gawdzik, B. Synthesis, Characterization and Testing of Antimicrobial Activity of Composites of Unsaturated Polyester Resins with Wood Flour and Silver Nanoparticles. Materials 202114, 1122. https://doi.org/10.3390/ma14051122] as well as information on regulations in application, e.g. European law. Silver can also be replaced with much more expensive gold and cheaper copper with equally well-known anti-microbial and less toxic properties.

Please expand Introduction, and Conclusions sections.

Please expand discussion of results.

Please provide information that distinguishes the presented research from the solutions known in science.

The work should be significantly improved so that it can be analyzed more thoroughly.

Best reagrds,

Author Response

(The authors gave the same response as above.)

Reviewer 3 Report

Overall, this manuscript demonstrates the functionalization of PET using ethylenediamine, sliver nanoparticles and dopamine, showing improved antibacterial and antioxidant performance. The fabrication is simple and practical, and the experimental data supports the design of the material. This work is worthy of publication in Polymers. However, the following concerns should be addressed before publication. In addition, the editing of the manuscript should be greatly improved.

Major concerns:

1. The description of y-axis in Figure 4 (thermograms) is wrong. For TGA, it should be “Weight %” or “Mass %”, not “Weight loss %”. They are the opposite results. In addition, the description on TGA results is too simple. Since there are three phases observed, what are those phases? Please clarify.

2. I understand that the second figure in Figure 2 is the zoom-in description, however, it is not appropriate to show the data like this (its repetitive data). If the authors want to show the details, please use inset figure and clarify in the figure legend.

3. In Figure 8, the antibacterial effect on Staph. aureus was not much different between PET-EDA and AgNPs-loaded PET-EDA, while the effects was obvious in E coli case. Is there any explanation? Is it because that Staph. aureus is more resistant to AgNPs?

Minor concerns:

1.     Please check all the abbreviations in the paper. Please add the full chemical name of PET when first introduce the concept in the paper; please also indicate the full name of EDA,FESEM and DPPH, etc.

2.     Please revise the figures (e.g., Figure3, there is part of the figure missing in Figure 3d; the figure legend is too simple, all the four panels should be described).

3.     Please carefully check the spelling and grammar, there are several errors. E.g., Page2, Line57, “This consisted of also of polyester…”. In addition, why are there some quotations in the experimental section (e.g., Page2, Line 61, 69, and 78)? If the authors are citing the previously reported methods, corresponding references should be added.

4.     Please carefully check all the units used in the manuscript and keep the format identical.

Author Response

(The authors gave the same response as above.)

Round 2

Reviewer 1 Report

The paper was essentially improved after revision but some minor issues remain yet. 

What does the sentence "The advantage of this work is attributed to the ethylene unit supporting crosslinking between polymeric chains" means? It is completely unclear to me. This information should be strongly expanded. Other advantages also should be added. 

Please correct the information about reference 6 where a number of the volume is absent. Now is  "Chemical Engineering Journal, 137048." Rightly "Chemical Engineering Journal, 446, 137048."

Author Response

R1

The paper was essentially improved after revision, but some minor issues remain yet. 

What does the sentence "The advantage of this work is attributed to the ethylene unit supporting crosslinking between polymeric chains" mean? It is completely unclear to me. This information should be strongly expanded. Other advantages also should be added. 

response: This sentence means that the treatment with ethylene diamine has the advantage of activating the PET fabric with amidoethylene amine, which would render the fabric basicity and/or crosslinking possibility with adjacent chains. Furthermore, the mimicking structural amidoethylene amine unit with the replaced ethylene glycol unit in the PET fabric could lead to activated fabric with interesting potential characteristics.

Please correct the information about reference 6 where a number of the volume is absent. Now is "Chemical Engineering Journal, 137048." Rightly "Chemical Engineering Journal, 446, 137048."

response: Thank you for your nice suggestion. This reference was corrected.

R2

Comments and Suggestions for Authors

Dear Authors,

Thank you very much for considering my suggestions. I believe that the revised manuscript as it stands is fit for publication.

Best wishes

Response: We would like to thank the Reviewer for carefully reviewing the manuscript and providing comments and suggestions to improve the quality of the manuscript.

R3

Comments and Suggestions for Authors

The revised manuscript has already addressed the concerns in the previous review comments. The description on the data matches the experimental design. In addition, more details have been added in the results section, which provides a more comprehensive discussion. Overall, the revised manuscript is suitable for publication in Polymers.

Response: We would like to thank the Reviewer for carefully reviewing the manuscript and providing comments and suggestions to improve the quality of the manuscript.

Reviewer 2 Report

Dear Authors,

Thank you very much for considering my suggestions. I believe that the revised manuscript as it stands is fit for publication.

Best wishes

Author Response

Dear Authors,

Thank you very much for considering my suggestions. I believe that the revised manuscript as it stands is fit for publication.

Best wishes

Response: We would like to thank the Reviewer for carefully reviewing the manuscript and providing comments and suggestions to improve the quality of the manuscript.

Reviewer 3 Report

The revised the manuscript has already addressed the concerns in the previous review comments. The description on the data matches the experimental design. In addition, more details have been added in the results section, which provides more comprehensive discussion. Overall, the revised manuscript is suitable for publication in Polymers.

Author Response

Comments and Suggestions for Authors

The revised manuscript has already addressed the concerns in the previous review comments. The description on the data matches the experimental design. In addition, more details have been added in the results section, which provides a more comprehensive discussion. Overall, the revised manuscript is suitable for publication in Polymers.

Response: We would like to thank the Reviewer for carefully reviewing the manuscript and providing comments and suggestions to improve the quality of the manuscript.
